# Identification of a Tumor Cell Associated Type I IFN Resistance Gene Expression Signature of Human Melanoma, the Components of Which Have a Predictive Potential for Immunotherapy

**DOI:** 10.3390/ijms23052704

**Published:** 2022-02-28

**Authors:** Andrea Ladányi, Erzsébet Rásó, Tamás Barbai, Laura Vízkeleti, László G. Puskás, Szonja A. Kovács, Balázs Győrffy, József Tímár

**Affiliations:** 1Department of Surgical and Molecular Pathology, National Institute of Oncology, 1122 Budapest, Hungary; ladanyi.andrea@oncol.hu; 22nd Department of Pathology, Semmelweis University, 1091 Budapest, Hungary; rasoerzs@gmail.com (E.R.); tbarbai@gmail.com (T.B.); laura.vizkeleti@gmail.com (L.V.); 3Avidin Ltd., 6726 Szeged, Hungary; laszlo@avidinbiotech.com; 4Department of Bioinformatics, Semmelweis University, 1094 Budapest, Hungary; kovacs.szonja@phd.semmelweis.hu (S.A.K.); gyorffy.balazs@yahoo.com (B.G.); 5Research Centre for Natural Sciences, Oncology Biomarkers Research Group, Institute of Enzymology, Eötvös Loránd Research Network, 1117 Budapest, Hungary

**Keywords:** human melanoma, preclinical model, type I interferon resistance, gene expression, immunotherapy resistance

## Abstract

We developed a human melanoma model using the HT168-M1 cell line to induce IFN-α2 resistance in vitro (HT168-M1res), which was proven to be maintained in vivo in SCID mice. Comparing the mRNA profile of in vitro cultured HT168-M1res cells to its sensitive counterpart, we found 79 differentially expressed genes (DEGs). We found that only a 13-gene core of the DEGs was stable in vitro and only a 4-gene core was stable in vivo. Using an in silico cohort of IFN-treated melanoma tissues, we validated a differentially expressed 9-gene core of the DEGs. Furthermore, using an in silico cohort of immune checkpoint inhibitor (ICI)-treated melanoma tissues, we tested the predictive power of the DEGs for the response rate. Analysis of the top four upregulated and top four downregulated genes of the DEGs identified *WFDC1*, *EFNA3*, *DDX10*, and *PTBP1* as predictive genes, and analysis of the “stable” genes of DEGs for predictive potential of ICI response revealed another 13 genes, out of which *CDCA4*, *SOX4*, *DEK*, and *HSPA1B* were identified as IFN-regulated genes. Interestingly, the IFN treatment associated genes and the ICI-therapy predictive genes overlapped by three genes: *WFDC1*, *BCAN*, and *MT2A*, suggesting a connection between the two biological processes.

## 1. Introduction

Type I and type II IFNs are cytokines primarily produced by virus-infected cells to initiate innate immune responses. Although they share many biological functions, they have distinct receptors. Type I IFNs signal through IFNAR1/2 heterodimeric receptors, activate Tyk2 and JAK1, and phosphorylate STAT1/2, which form complexes with IRF9, translocate to the nucleus, and activate expression of interferon-stimulated genes (ISG). Type II IFN (IFN-γ) also has a heterodimeric receptor, IFNGR1/2, which, upon ligand binding, activates JAK1/2 and phosphorylates STAT1, resulting in nuclear translocation and activation of the ISGs. However, IFNRG1/2 can activate alternative signaling pathways as well (STAT4, ERK1/2, Pyk2, or CRK1) [1].

There are over 2000 IFN-regulated genes (IRGs) known today, including growth factors (like VEGF, FGF, and ECGF), chemokines (such as MIB, EBI1, and IL-8), adhesion molecules (i.e., ICAM1, CD47, and ALCAM), MHC class I and II, apoptosis regulators (such as FAS and CASP4/8), signaling molecules (like IFI16 and STAT1/2), and several transcription factors (including IRF1-7, ISGF3G, MPB1, PBX3, and, interestingly, HIF1α). A comprehensive database of interferon-regulated genes is available on the Interferome website [2].

Cytokine therapy of melanoma patients has a long history. As malignant melanoma is relatively resistant to chemotherapy and radiotherapy, for a long time, cytokine therapy was the option in adjuvant settings using type I IFN monotherapy or in combination with type II IFN or IL-2 [3]. However, IFN therapy has very low efficacy: ~6% for 5-year disease-free survival and 3% for 5-year overall survival according to major recent meta-analyses [4]. This low efficacy is improved in the case of high-risk ulcerated primary tumors, but disappears in the case of non-ulcerated ones [3,4]. Accordingly, malignant skin melanoma can be considered “by default” to be IFN resistant, although the contribution of the tumor or the host to this feature is not known. Accordingly, the type I IFN sensitivity/resistance issue in the case of melanoma was open for a long time. 

Pioneer studies on unselected human melanoma cell lines identified the RCC1, IFI16, HOX2, and H19 signature of sensitivity and SHB and PKCζ as markers of resistance to type I IFN [5]. Another study on human melanoma xenografts identified a five-IRG signature of sensitivity comprising MxB, leu-13, Kip1/p27, Rig-E, and BST-2 [6]. Type I IFN signaling is involved in the regulation of oncogene-induced senescence [7], therefore it was studied in BRAF-mutant melanocytes and melanomas in animal models. The role of type I IFNs in the carcinogenesis and the progression of BRAF-mutant melanoma was analyzed in a genetically manipulated mouse model where IFNAR1 was knocked out [7]. Data indicated that melanoma carcinogenesis was promoted in the IFNAR1 deficient host and the resulting tumors were spontaneously metastatic. Restoration of IFNAR1 signaling in BRAF mutant melanoma cells attenuated tumor growth in vivo. However, analysis of IFN-treated human melanoma tumor samples showed only a trend of lower IFNAR1 with a poorer outcome of patients, suggesting that the host may also participate in the regulation of IFN signaling [7,8]. 

Immune checkpoint inhibitor (ICI) therapy fundamentally changed the management of melanoma patients, significantly improving the five-year survival [9]. However, there is no useful predictive marker of efficacy and data indicate that this treatment does not work in ~50% of patients [9,10]. Predictive markers of ICI therapy in solid tumors have been developed and are widely used today [11]. Unfortunately, in melanoma, the predictive role of the PD-L1 expression is controversial [12], microsatellite instability is very rare [13], and only a high tumor mutational burden is clinically relevant [14]. Clinical research revealed that type II IFN signaling is necessary for efficacy of anti-CTLA-4 and anti-PD-1 immunotherapy [15]. Another analysis revealed that an established IFN-γ gene expression signature score, together with the tumor mutation burden, is a powerful predictor of the efficacy of immunotherapy of melanoma patients [16]. Using experimental melanoma models and anti-PD-1 treated patient-derived tumor tissues, it was revealed that maintained type I IFN signaling is also a necessary element of the efficacy of immunotherapy [8]. Analysis of the gene expression of melanoma tissues during CTLA-4 and high dose IFN-α2b combination therapy revealed a pro-inflammatory gene signature as a predictor of response and efficacy [17]. These data all point to the importance of type I IFN signaling in ICI therapy response, but do not differentiate between the stromal and tumor components. In the case of type II IFN signaling, tumor intrinsic responses have been observed, mainly consisting of WNT and MYC signaling component, as well as the components of the antigen presentation machinery [15]. A study performed on a large melanoma database treated with anti-CTLA-4 therapy revealed that homozygous deletion of type I IFN genes is significantly associated with resistance [18], further supporting the notion that type I IFN signaling of tumor cells may play a significant role in melanoma progression. 

## 2. Results

### 2.1. Selection of IFN-α-Resistant Melanoma Cell Line

HT168-M1 melanoma cells, grown as monolayer cultures, were cultured for 6 weeks in the presence of escalating doses (10,000 U/mL, then 20,000 U/mL) of IFN-α2a, followed by testing for sensitivity in vitro treatment with IFN-α2a using the MTT assay. IFN-α2a treatment resulted in a concentration-dependent growth inhibition of 55–80% of the parental cells, while no significant effect was found in the case of the selected line (termed HT168-M1res; Figure 1). Following selection, the HT168-M1res cell line was cultured in a regular medium in the absence of IFN-α. Regular testing in the MTT assay proved that it maintained resistance to IFN-α2a (data not shown).

### 2.2. Effect of In Vivo IFN-α2a Treatment on Growth of HT168-M1 and HT168-M1res Tumors after Intrasplenic Injection into SCID Mice

We also tested the sensitivity of the parental HT168-M1 line and its selected variant on the effect of in vivo treatment with IFN-α2a after intrasplenic tumor cell injection in SCID mice. The animals were treated intraperitoneally with IFN-α2a daily, six times a week for 16 days, starting 2 days after tumor cell injection. Interferon treatment significantly reduced primary tumor growth in mice injected with HT168-M1 cells, resulting in a 69% and 86% decrease in the weight of the splenic tumor in mice treated with 10^5^ and 5 × 10^5^ U IFN-α2a, respectively (Figure 2). In the case of the HT168-M1res line, a smaller and statistically not significant decrease was observed (Figure 2). No significant effect was found on the number of liver metastases in either of the melanoma lines studied (data not shown). To test how stable this resistance was, short-term (1 week) cultures derived from the primary tumors (HT168-M1 and HT168-M1res) of IFN-α2a-treated and control mice were tested for sensitivity to IFN-α2a in in vitro proliferation assays. The results showed that both cell lines retained their sensitivity characteristics, regardless of the in vivo treatment applied (Appendix A).

### 2.3. Identification of Genes Differentially Expressed in IFN Resistant and IFN Sensitive Melanoma Cells

Using in vitro cultured HT168-M1 and HT168-M1res cells, we compared the mRNA expression profiles. This analysis revealed 91 genes significantly differently expressed in the resistant melanoma cells. Ten genes were not present in the repeated experiment and another two were lncRNAs, in this way, we obtained 79 differentially expressed genes (Appendix A). Of these DEGs, 24 belonged to interferon-regulated genes (IRGs) according to the Interferome portal [2] (Table 1): 14 IRGs were upregulated, while 10 were downregulated. The non-IRG genes consisted of 55 genes, of which 33 were upregulated and 22 were downregulated (Appendix A); the top 10 genes of each category are shown in Table 2. 

PANTHER gene ontology analysis (www.pantherdb.org, 8 April 2021) of the IRGs identified IFN signaling exclusively as a significantly represented pathway (FDR-corrected *p* = 0.0035). However, analysis of the 55 non-IRGs identified 14 biological processes, where the 5 most significant ones are presented in Table 3. Regulation of Ca-signaling had the highest significance and FDR values, while the four others all belonged to neuronal development.

### 2.4. Testing the In Vitro/In Vivo Stability of the Differentially Expressed Genes Using TaqMan Assay

We further tested the stability of the DEGs of the in vitro cultured HT168-M1res cells and found a 13-gene core signature that maintained the same direction of differences as was originally observed (Table 4 and Appendix A). This 13-core DEG contained four IRGs: *SOX4*, *UCP3*, *DEK*, and *HSPA1B*.

When testing the DEGs obtained using in vitro cell lines in in vivo growing tumors, in the first experiment, HT168-M1res tumors differentially expressed 23 genes (9 up- and 14 down-regulated; Appendix A). In a repeated in vivo experiment, a smaller set of 19 DEGs was obtained (4 up- and 15 down-regulated; Appendix A). Meanwhile, only four genes were present in the two in vivo expression runs from the original DEGs obtained comparing the in vitro cultured, IFN resistant, and sensitive cells (Appendix A). This four-gene core IFN-resistance signature contained two IRGs, *IFI27* and *CDCA4*, and two non-IRGs, *CDKL3* and *AQP1*.

### 2.5. Analysis of TCGA mRNA Datasets of IFN-Treated Melanoma Metastases

The expression of individual components of the IFN resistance DEGs (Appendix A) was analyzed on melanoma samples extracted from the TCGA database. First, we filtered for metastatic tissues with known prior systemic therapy data for their primary tumor (Cohort 1, *n* = 33; Appendix A). Four genes out of the 79 DEGs were found to be significantly differently expressed in this sample cohort. *WFDC1* was significantly downregulated, whereas *BCAN*, *SOX4*, and *RPE65* were upregulated in metastasis with prior IFN therapy (*n* = 27) compared to the any other treated (*n* = 6) samples (Table 5). *HOXC11* showed a trend of upregulation in the IFN-treated cases.

We also analyzed metastatic samples with known systemic therapy for the uploaded metastatic tissue (Cohort 2, *n* = 69; Appendix A). Tumors treated with immunotherapeutic agents (both alone and in combination therapy) other than IFN were excluded from this analysis. Downregulation of *MAPT* in the IFN-treated (*n* = 18) group compared to the other treatments (*n* = 51) was the solely significantly altered gene from our DEGs (fold change: 0.442, *p* = 0.027, q = 0.027). 

Furthermore, we tested the potential predictive role of the IFN resistance DEGs for interferon therapy. TCGA contained only a limited number of samples from patients with a known therapy response after IFN treatment for progressive or stable disease (*n* = 4) and complete or partial response (*n* = 4). However, samples with an unfavorable therapy response were characterized by significant downregulation of *HSPB7*, *MT2A*, *HSF1*, *WFDC1*, and *TPD52L1* (nearly significant), while the marginal upregulation of *DOCK11*, *DEK*, and *SOX4* only approached significance (Table 6).

### 2.6. Testing the Predictive Role of Differentially Expressed Genes in the Immune Checkpoint Inhibitor Therapy of Melanoma Patients

When searching for transcriptomic datasets with immune checkpoint inhibitor (ICI) treatment in cutaneous melanoma patients, we uncovered six eligible datasets. The number of samples with documented ICI treatment responses eligible for our study was 318, but due to platform differences, not all datasets had data for all genes. In each of these cases, sample acquisition was performed before the administration of ICI treatment. The administered treatments included anti-PD-1 (*n* = 228) and anti-CTLA-4 (*n* = 48) antibodies, and a small cohort received both drugs (*n* = 42). Some patients received multiple treatment regimens (Appendix A).

When using the published response data, 196 patients were non-responders and 122 were responders. The analysis was performed using author-reported response data as the end-points defining resistance and sensitivity. The statistical analysis was performed by computing a Mann−Whitney U-test. We selected the top four upregulated (*WFDC1*, *GAGE2C*, *EFNA3*, and *DDX10*) and the top four downregulated genes (*CAMK1*, *PTBP1*, *TNFSF10*, and *SLC17A3*) of the 79 DEGs (Appendix A) and analyzed their individual predictive power of ICI response. Out of the eight genes, four were found to be expressed significantly differently in responders (Table 7)—*WFDC1*, *EFNA3*, and *PTBP1* were downregulated, while *DDX10* was upregulated in responder patients. It is of note that all of these genes belonged to the non-IRG gene category.

We also tested the ICI-therapy predictive power of the in vitro gene core of IFN resistance DEGs (Table 4), and found that 11 out of 13 genes were expressed significantly differently in non-responsive melanomas (Table 8). It is of note that three genes out of the 11 belonged to IRGs (*SOX4*, *DEK*, and *HSPA1B*).

Furthermore, we tested the individual predictive power of the mRNA expression of the in vivo selected DEGs for ICI treatment response in this patient cohort. Statistical analysis indicated that *AQP1* and *CDCA4* were also downregulated in responder patients (Table 9).

## 3. Discussion

Here, we report an experimental model where a novel IFN-α2-resistance gene expression signature (GES) was defined. IFN resistance was developed in vitro by long-term exposure to type I IFN of human melanoma cells HT168-M1. This IFN resistance was maintained in vivo when tumor cells were inoculated into SCID mice. The development of this resistance mechanism was independent of immune mechanisms, because it was developed during in vitro culturing and was maintained in immune suppressed rodent hosts; accordingly, it was intrinsic of melanoma cells. Using microarray analysis of the in vitro cultured melanoma cells, we defined 79 differentially expressed genes. Of these 79 genes, 24 belonged to IFN-regulated genes according to the Interferome analysis [2]. Accordingly, the majority of the identified melanoma-related genes were not IFN-regulated. Clinical studies have indicated that upon recurrence after IFN-α therapy, melanomas overexpress STAT5 [19], but this gene was not part of our list of differentially expressed genes. Similarly, none of the previously identified IFN resistance genes were present in this signature [5,6], most probably due to the non-immune mechanism of the development of resistance. 

The PANTHER pathway analysis revealed that a significant component of the non-IRG part of DEGs belonged to the neuronal development pathways. Previous studies revealed that during melanoma progression, melanoma cells develop stem-cell like properties associated with the expression of SOX10, EZH2 transcription factors, EMT phenotypic switch regulators TWIST1/ZEB1, and the surface receptor CD172 [20]. In the non-IRG gene list of IFN resistance, we found the *TYRP1* melanoma marker, *SSTR5* somatostatin receptor, and *RPE65* retinal pigment epithelial marker genes all overexpressed in resistant melanoma cells, suggesting that neural crest and melanocytic linage markers may have a role in developing IFN resistance. Furthermore, a recent analysis of melanomas exposed to anti-PD-1 therapy revealed alterations in the expression of melanocytic and neural crest-related genes [21]. 

Our in silico analysis of IFN-treated melanoma tissues of TCGA revealed the differential expression of five members of our 79 IFN-res DEGs in IFN-treated melanoma tissues: IRG *SOX4* and non-IRGs *WFDC1*, *BCAN*, *RPE65*, and *MAPT*. The SOX4 transcription factor is reported to be upregulated in melanoma believed to be involved in metabolic rewiring [22]. WFDC1 is a tumor suppressor frequently lost in breast and prostate cancers, hepatocellular carcinoma, and Wilms’ tumor. In a significant proportion of melanomas, WFDC1 is downregulated by hypermethylation and has been shown to inhibit expression of DKK1, a known WNT signaling inhibitor [23]. *DKK1* is part of the non-IRG DEGs, where it is significantly downregulated compared to *WFDC1*, which is the most upregulated one. It is of note that in our experimental IFN resistance models, *WFDC1* was consistently found to be differentially expressed in IFN resistant and sensitive melanoma cells or tumors. BCAN (brevican) is a chondroitin sulphate proteoglycan of the ECM, with no data on its role in melanoma. RPE65 was shown to be expressed by nevi, but downregulated in melanoma [24]. The upregulated HOXC11 is also a transcription factor, regulating the expression of linage marker S100b in melanoma [25]. *MAPT* codes for the tau protein involved in Alzheimer and Parkinson’s diseases. Interestingly, recently, a connection between neurodegenerative diseases and melanoma was raised, demonstrating an accumulation of amyloid in melanoma metastases [26]. 

Although the IFN-treated TCGA cohort of melanoma was very small, having clinical response data only for eight patients, we also tested the predictive power of components of our IFN-res DEGs. This analysis revealed four genes expressed significantly differently in responder patients compared to nonresponders: non-IRGs *WFDC1*, *HSBP7*, *HSF1*, and the only IRG, *MT2A*. It is of note that marker genes of IFN therapy and predictive genes of IFN therapy efficacy only overlapped by one gene, *WFDC1*. This IFN therapy predictive gene set contained two members of the heat shock protein family, *HSPB7* and *HSF1*. While there were no data on the role of HSPB7 in melanoma, HSF1 was shown to be upregulated in melanoma due to the loss of FBX7, and was shown to be involved in regulating the metastatic potential [27]. The nearly significant TDP52L is a regulator of MAP3K5 protein kinase and has been shown to be involved in cell proliferation of melanoma cells [28]. The IRG, MT2A is a metallothionein protein responsible for heavy metal ion detoxification. MT2A and other family members were shown to be overexpressed in melanoma and were associated with increased macrophage density of TME [29].

Previous studies identified IFN signaling as the key predictive mechanism of the ICI therapies of melanoma. A tumor microenvironment-specific IFN-related ICI resistance signature was defined [30,31], as well as a tumor cell associated one [32]. The tumor-specific component of the ICI resistance was due to genetic loss of *IFNGR1/2* and *JAK2*, and amplification of IFN signaling inhibitors *SOCS1* and *PIAS4* [32]. Other studies found downregulation or loss of heterozygosity of *HLA-B* [33], *B2M* [34], *JAK1* [35], and *SERPINB3/4* mutations [36] as markers of ICI resistance. However, a recent analysis defined a predictive 30-gene IFN-γ pathway expression signature of anti-CTLA-4 therapy, which contained three genes of the IRG component of our DEGs—*CAMK2D*, *MT2A*, and *HSP90AB1* [37]. 

Analysis of the four most upregulated and four most downregulated genes of the in vitro obtained DEGs for their predictive power for response to ICI treatment identified four genes—*WFDC1*, *EFNA3*, *DDX10*, and *PTBP1*—with a significant predictive potential, and none of them were IRGs. We also analyzed the stability of these differentially expressed genes in vitro and in vivo using melanoma xenografts. We found that only a 13-gene subset of IFN resistance DEGs were stable in vitro, containing IRGs *SOX4*, *UCP3*, *DEK*, and *HSPA1B*. Furthermore, in two independent studies, we found that only a small subset of genes of the DEGs was present in vivo in xenografts consisting of a 4-gene core containing IRGs *IFI27* and *CDCA4* and two non-IRGs *AQP1* and *CDKL3*. When the predictive power of all of these genes was tested on an ICI-treated melanoma patient cohort, 11 out of the 13 in vitro stable (containing IRGs *SOX4*, *DEK*, and *HSPA1B*) and *AQP1* and *CDCA*4 in vivo stable DEGs were differentially expressed in the tumors of ICI responder melanoma patients. In this way, we defined a 17-gene core the IFN resistance DEGs of melanoma, which all had a predictive potential for the ICI response of melanoma patients (Table 7, Table 8 and Table 9).

*EFNA3* is a hypoxia regulated gene and is a GPI-anchored ligand for the EPH receptors involved in cell adhesion and motility. It is a negative prognostic factor of gastric, ovarian, and lung cancers [38,39,40]. DDX10 is an RNA helicase that is frequently lost in ovarian cancer [41] and is a poor prognostic factor in osteosarcoma [42]. PTBP1 is an RNA-binding protein involved in splicing. In dendritic cells, it was found that PTBP1 regulates the expression of several IFN-regulated genes [43]. PTBP1 is expressed by melanoma stem cells [44] and it has been shown to regulate CD44v6 expression in melanoma brain metastases [45]. CDCA4 is an interferon-regulated E2F-type transcription factor involved in cell cycle regulation. In melanoma, miR-15a and miR-29c-3p are regulators of CDCA4, which is involved in controlling cell proliferation, invasion, and apoptosis [46,47]. AQP1 is a water channel protein, a hypoxia-regulated gene involved in various biological processes. AQP1 was found to be overexpressed in BRAF-mutant melanoma tumors and was shown to be a negative prognostic factor [48]. 

It is of note that the in vitro stable IFN resistance 11-core DEGs contained two melanoma oncogenes, *SOX4* and *DEK* (transcription factors); the former was reported to be downregulated, while the latter was found to be upregulated during melanoma progression [49,50]. Furthermore, this core-DEG also contained two neuronal genes, *NPTXR* and *SSTR5*, suggesting that melanoma stem cell properties might also have a role in ICI therapy resistance [21]. Last, but not least, this core-DEG also contained two proteoglycans, BCAN (the chondroitin sulphate proteoglycan brevican) and SDC2 (the heparan sulphate proteoglycan, syndecan2). SCD2/syndecan2 was shown to be involved in the regulation of the migratory potential of melanoma cells [51]. Furthermore, recently, proteoglycans were shown to be involved in the adaptive immune escape of experimental melanoma, where GUSB glucuronidase plays a role as a novel oncosuppressor [52]. 

What could be the connection between type I IFN resistance and response to ICI therapy in melanoma? Type I IFN therapy was and is still is part of the management of melanoma patients [3,4]. From this perspective, it is of note that the IFN therapy and the ICI therapy predictive genes of our IFN resistance DEGs overlapped by three genes, *WFDC1*, *SOX4*, and *BCAN*, strongly suggesting a connection between the two pheno-/geno-types of melanoma: IFN and ICI resistance. Progression of the disease after IFN therapy could be interpreted as development of IFN resistance in treated patients. It is tempting to speculate that melanomas that progressed after type I IFN therapy may respond differently to ICI therapies compared to those where such a therapy was not administered previously. Genetic analysis of the progressed melanomas previously treated with type I IFN compared to those who were not could reveal such a possible connection.

## 4. Materials and Methods

### 4.1. Tumor Cells and Culture Conditions 

The HT168-M1 human melanoma cell line, a derivative of the A2058 line, was developed in our laboratory by in vivo selection for its high liver colonizing capacity [53]. Cells were maintained in vitro as monolayer cultures in the RPMI 1640 medium (Sigma, St. Louis, MO, USA) supplemented with 5% fetal calf serum (Sigma) and 50 μg/mL gentamycin at 37 °C in a 5% CO_2_ atmosphere. In vitro selection for IFN-α resistance was carried out by culturing the cells for 6 weeks in the presence of escalating doses (10,000 U/mL for 1 month, then 20,000 U/mL for 11 days) of IFN-α2a (Roferon-A, F. Hoffmann-La Roche, Basel, Switzerland) for 6 weeks. Following this period, the selected cell line (HT168-M1res) was maintained in the absence of IFN-α, and was regularly tested for IFN-α sensitivity.

### 4.2. Cell Proliferation

One thousand cells were plated in 96-well tissue culture plates, and after 24 h were treated with IFN-α2a (Roferon-A, F. Hoffmann-La Roche) at different concentrations. Cells were incubated for 120 h, then the relative cell density was determined by the MTT assay. Briefly, 0.5 mg/mL of the tetrazolium dye MTT (Sigma) was added to the wells, then after 4 h of incubation at 37 °C, the medium was gently removed, the plates air-dried, and the formazan crystals, formed in viable cells, were dissolved in DMSO. The absorbance at 570 nm was measured with a Bio-Rad microplate reader (Hercules, CA, USA).

### 4.3. Experimental Animals

SCID (CB17/Icr-*Prkdc^scid^*) mice were obtained from Charles River Laboratories (Wilmington, MA, USA), and bred and housed in the pathogen-free animal facility of the National Institute of Oncology, Budapest. All animal studies were conducted in accordance with published guidelines on the welfare of animals in cancer research [54]. The experimental protocols were approved by the Animal Care and Use Committee of National Institute of Oncology.

### 4.4. In Vivo Treatment of Human Melanoma Tumors after Intrasplenic Tumor Cell Injection 

HT168-M1 and HT168-M1res cells (5 × 10^4^/mouse) were injected into the spleen of female SCID mice at a volume of 50 μL. The mice were treated intraperitoneally with IFN-α2a at doses of 10^5^ and 5 × 10^5^ U in 100 μL physiological saline daily six times a week for 16 days, starting on day 2 after tumor cell injection. The control animals received physiological saline only. The experiments were terminated 30 days after tumor cell inoculation, the spleen (primary tumor) and liver of the animals were weighed and fixed in 4% formalin, and liver surface colonies were counted under a stereomicroscope. The experiment was repeated twice and the results of a representative experiment are shown. At termination, one primary tumor of each treatment group was used for producing tissue cultures, and 7-day-old cultures were tested for sensitivity to in vitro treatment with IFN-α2a using the MTT test, as described above.

For statistical evaluation of the results for the in vitro experiments, Student’s t-test was used, while those for the in vivo experiments were analyzed with the Mann−Whitney U-test.

### 4.5. RNA Preparation

The total RNA was isolated from the frozen samples of two different cultures of HT168-M1 and HT168-M1res cells. The total RNA was isolated from the frozen homogenized samples using the RNeasy Mini kit (Qiagen, Hilden, Germany) according to the manufacturer instructions. Possible DNA contamination was eliminated using RNase-free DNase Set (Qiagen) using on-column DNA digestion in the RNeasy mini kit protocol. The RNA integrity was checked by denaturing agarose gel electrophoresis on a 1% agarose gel containing 12.3 M formaldehyde and a MOPS running buffer. DNA contamination was excluded by carrying out a PCR reaction with β-actin primers using RNA samples as the templates. In the same reaction using common β-actin PCR master mix, positive controls were the reverse transcribed cDNA of the same samples. The quality and quantity of isolated RNA was checked by electrophoresis and spectrophotometry (NanoDrop, Rockland, DE, USA).

### 4.6. Microarray Hybridization and Quantitative PCR Validation

RNA samples were hybridized to the Human Genome Survey Microarray V2.0 (Applied Biosystems, Waltham, MA, USA) containing 32,878 probes. Arrays were scanned by AB1700. Data of <3 S/*n* were eliminated from the database, which was then analyzed by GeneSpring and a list of 385 genes was produced that were differentially expressed (>2-fold up- or downregulation). A repeated analysis still contained a large set of 335 genes, therefore it was downsized by using lower *p* values (0.001), resulting in a gene list of 91 genes. The raw microarray data are available as Appendix A while submission into the Gene Expression Omnibus repository is pending.

To validate the IFN resistance gene expression signature obtained in vitro, RNA was isolated from repeated in vitro cultures and subdermal xenograft tumor tissues of HT168-M1 and HT168-M1res cells and tumors (three to four samples/group). For quantitative measurement of the gene expression, a TaqMan Low Density Array (Applied Biosystems) containing the 91 genes obtained in vitro was used. The q-PCR reaction mixture of 25 μL contained 12.5 μL of 2 × SYBR Green Supermix (Bio-Rad), 0.5 μL of the individual primers for final concentration of 200 nM, and 11.5 μL of diluted cDNA. The cycling conditions were 3 min iTaq DNA polymerase activation at 95 °C, 40 cycles at 95 °C for 30 s, at 55 °C for 30 s, and at 72 °C for 1 min. Starting quantities were defined by standard five-fold dilution series carried out with control cDNA of human K562 cells. Relative expression of the examined genes was determined by normalizing the starting quantities to those of the housekeeping genes of the cDNA sample [55]. The results of the q-PCR were analyzed with Stat Soft Statistica11 software using unpaired *t*-test. 

### 4.7. Testing the Predictive Role of Differentially Expressed Genes

To evaluate the correlation between DEGs and response to immune checkpoint inhibition, we searched the GEO and EGA repositories to identify melanoma patient samples with available microarray-based or RNAseq-based gene expression data, and published the treatment information. Altogether, five datasets were identified (GSE91061, GSE78220, GSE115821, [56,57]). Gene annotation for the different datasets was performed by utilizing the HGNC defined gene symbols (www.genenames.org, Accessed on 21 October 2021) for each gene. All patients were assigned to responder and non-responder cohorts based on the author-reported clinical response data (Appendix A). The transcriptomic datasets were quantile normalized across all genes and the differential expression was determined using a Mann−Whitney U test, as described earlier [58], and can be found as Appendix A.

### 4.8. Analysis of RNA Data of TCGA Database 

RNA data of skin cutaneous melanoma tissue samples from a data portal filtered for metastatic samples (Cohort 1) with known prior systemic therapy (interferon (IFN, *n* = 27) versus other, *n* = 6) for primary tumor and (Cohort 2) with known therapy for the uploaded tissue data (IFN, *n* = 18 versus other, *n* = 51) were used for the gene expression analysis (Appendix A). In the latter cohort, applied immunotherapy other than IFN was excluded from the analysis. Normalized, level 3 data were downloaded and after log2 transformation were subjected to analysis using the Bioconductor BRB-ArrayTools 4.6.2 (Richard Simon and Amy Peng Lam, National Cancer Institute, Bethesda, MD, USA). Genes were excluded if (1) less than 20% of data have at least a 1.5-fold change in either direction from the gene’s median value, (2) the log-ratio variation was *p* > 0.05, or (3) at least 50% of data were missing or filtered out. We considered the alteration significant at the nominal 0.05 level using univariate tests (two-sample t-test, permutation test for significant genes using 10,000 random permutations) with at least a two-fold change.

## 5. Conclusions

Here, we described differentially expressed genes associated with IFN resistance for in vitro cultured human melanoma cells. Although these genes contained a significant proportion of IFN-regulated genes as expected, the majority of them were non-IRGs involved in various cellular processes. In vitro/in vivo analysis of the stability of this type I IFN resistance signature indicated that only a smaller proportion of the original gene set was expressed differentially in the repeated analyses, and only a 13-gene core was stable in vitro and a 4-gene core (*AQP1*, *CDKL3*, *IFI27*, and *CDCA4*) in vivo. Using an in silico cohort of IFN-treated melanoma tissues, we validated a differentially expressed 9-gene core of DEGs, out of which four genes had a predictive power for efficacy: *WFDC1*, *HSPB7*, *HSF1*, and *MT2A*. Evaluation of the predictive power of this novel signature in silico on an ICI-treated melanoma patient cohort extracted from GEO and EGA databases, revealed *WFDC1*, *EFNA3*, *DDX10*, and *PTBP1* as marker genes. Analysis of the individual elements of the stable signature of DEGs for predictive power revealed that 11 out of 13 core in vitro genes and *AQP1* and *CDCA4* of the in vivo core were also predictive marker genes. Importantly, the IFN treatment marker genes and the ICI therapy predictive genes overlapped by three genes (*WFDC1*, *BCAN*, and *MT2A*), suggesting a connection between the two biological processes—IFN- and ICI-therapy resistance.

## Figures and Tables

**Figure 1 ijms-23-02704-f001:**
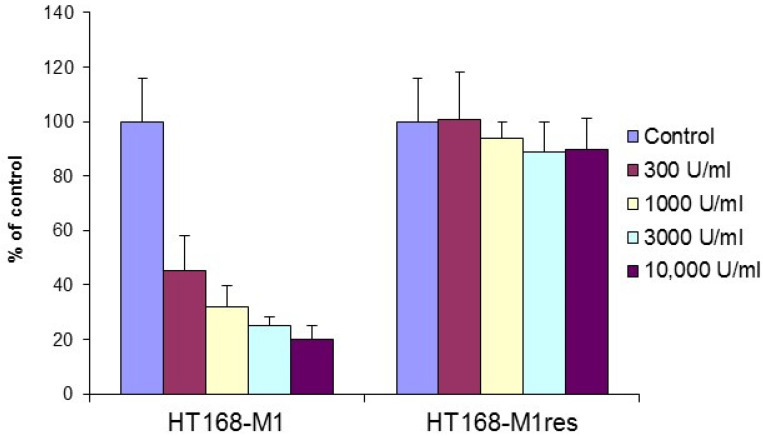
Effect of 5-day-long IFN-α2a treatment on the proliferation of HT168-M1 and HT168-M1res cells (MTT assay, six parallel samples, mean ± SD). *p* < 0.001 for all treatment concentrations vs. control in the case of HT168-M1, *p* > 0.05 in HT168-M1res cells (Student’s *t*-test).

**Figure 2 ijms-23-02704-f002:**
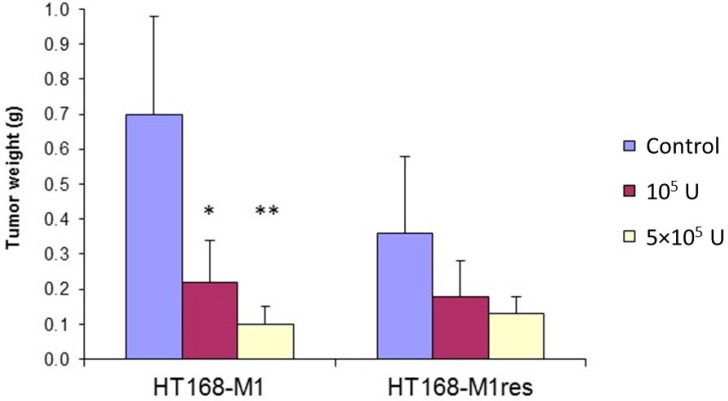
Effect of in vivo IFN-α2a treatment on primary tumor growth after intrasplenic injection of HT168-M1 and HT168-M1res cells (seven mice per group, mean ± SD; * *p* < 0.005, ** *p* < 0.001 compared to the control, using the Mann−Whitney U-test).

**Table 1 ijms-23-02704-t001:** IFN-regulated genes of the 79 DEGs in IFN resistant vs. IFN sensitive melanoma cells.

Fold Change	Gene Symbol	Gene Name	RefSeq
4.744	*TSPAN8*	Tetraspanin 8	NM_017955.4
3.2609	*MT2A*	Metallothionein 2A	NM_005532.5
3.071	*HSPA1B*	Heat shock 70 kDa protein 1B	NM_001085.5
2.79	*DOK5*	Docking protein 5	NM_004616.3
2.751	*DEK*	DEK proto-oncogene (DNA binding)	NM_001472.2
2.7291	*PDE1C*	Phosphodiesterase 1C, calmodulin-dependent 70 kDa	NM_001123067.3
2.651	*JDP2*	Jun dimerization protein 2	NM_033025.6
2.615	*EGR1*	Early growth response 1	NM_001379451.1
2.448	*ZNF703*	Zinc finger protein 703	NM_001053.3
2.181	*ATF5*	Activating transcription factor 5	NM_005978.4
2.17	*NDRG1*	N-myc downstream regulated gene 1	NM_005526.4
2.168	*CDCA4*	Cell division cycle associated 4	NM_000440.2
2.138	*CPXM1*	Carboxypeptidase X (M14 family), member 1	NM_014424.5
2.104	*CTSB*	Cathepsin B	NM_005953.5
0.486	*IFIT1*	Interferon-induced protein with tetratricopeptide repeats 1	NM_001191057.4
0.3617	*PAX3*	Paired box gene 3 (Waardenburg syndrome 1)	NM_001199264.23
0.347	*MX1*	MX dynamin like GTPase 1	NM_002819.5
0.3076	*SOX4*	SRY (sex determining region Y)-box 4	NM_006632.3
0.2714	*GPI*	Glucose phosphate isomerase	NM_003412.3
0.26	*SERPINA3*	Serpin family A member 3	NM_003447.4
0.211	*IFI27*	Interferon, alpha-inducible protein 27	NM_025069.2
0.192	*UCP3*	Uncoupling protein 3 (mitochondrial, proton carrier)	NM_019089.5
0.1572	*TNFSF10*	Tumor necrosis factor (ligand) superfamily, member 10	NM_012068.5
0.0721	*CAMK1*	Calcium/calmodulin-dependent protein kinase I	NM_019609.4

**Table 2 ijms-23-02704-t002:** Top 10 upregulated and downregulated non-IRGs in HT168-M1res cells compared to HT168-M1 cells.

Fold Change	Gene Symbol	Gene Name	RefSeq
15.21	*WFDC1*	WAP four-disulfide core domain 1	NM_012232.6
9.015	*GAGE2C*	G antigen 2	NR_026881
6.1534	*EFNA3*	Ephrin-A3; EFL2, EPLG3, Ehk1-L, HGNC:3223, LERK3	NM_001256374.1
5.3656	*DDX10*	DEAD (Asp-Glu-Ala-Asp) box polypeptide 10	NM_016508.4
4.963	*WNT7A*	WNT7A wingless-type MMTV integration site family, member 7A	NM_014212.
4.7162	*ABCC1*	ATP-binding cassette, sub-family C (CFTR/MRP), member 1	NM_144658.4
4.3842	*S100A2*	S100 calcium binding protein A2	NM_001548.5
4.298	*PTRF/CAVIN1*	Polymerase I and transcript release factor	NM_015558
3.979	*PRG1*	Proteoglycan 1, secretory granule	NM_053039.2
3.8076	*HSF1*	Heat shock transcription factor 1	NM_001290060.2
0.281	*PHACTR1*	Phosphatase and actin regulator 1	NM_018337.4
0.2567	*EZF-2 (ZNF444)*	Zinc finger protein 444	NM_004398.3
0.2567	*ZIC1*	Zic family member 1 (odd-paired homolog, Drosophila)	NM_000550.2
0.2372	*HDAC8*	Histone deacetylase 8	NM_018431.5
0.231	*LRRK2*	Leucine-rich repeat kinase 2	NM_001135047.2
0.2225	*WT1*	Wilms tumor 1	NM_001964.2
0.2165	*FGF20*	Fibroblast growth factor 20	NM_002148.3
0.195	*DKK1*	Dickkopf WNT signaling pathway inhibitor 1	NM_002199.3
0.1864	*SLC17A3*	Solute carrier family 17 (sodium phosphate), member 3	NM_024505.3
0.1023	*PTBP1*	Polypyrimidine tract binding protein 1	NM_006096.3

**Table 3 ijms-23-02704-t003:** PANTHER overrepresentation test of the 55-gene non-IRG component of the 79 DEGs.

GO Biological Process	GO Genes	Non-IRG Genes	Over/Under	Fold Enrichment	*p*-Value	FDR
calcium signaling (GO:0050848)	68	4	0.18	21.63	4.35 × 10^−5^	4.90 × 10^−2^
generation of neurons (GO:0048699)	1249	13	3.4	3.83	2.50 × 10^−5^	3.03 × 10^−2^
neuron differentiation (GO:0030182)	1028	12	2.8	4.29	1.79 × 10^−5^	2.35 × 10^−2^
neuron projection development (GO:0031175)	683	10	1.86	5.38	1.50 × 10^−5^	2.15 × 10^−2^
neuron development (GO:0048666)	833	11	2.27	4.86	1.38 × 10^−5^	2.17 × 10^−2^

Annotation version and release date: GO Ontology database DOI:10.5281/zenodo.4495804 (released 1 February 2021). Reference List: homo sapiens (all genes in database), test type: Fisher exact, FDR, *p* value was tested by Mann−Whitney U test. Data are expressed as the normalized gene expression.

**Table 4 ijms-23-02704-t004:** In vitro stable DEGs of HT168-M1res melanoma cells.

Fold Change	Gene Symbol	Gene Name	RefSeq
5.769	*PDE6A*	Phosphodiestherase 6A	NM_000440.1
2.567	*EFHD1*	EF-hand domain family member D1	NM_030948.6
2.441	*BCORL1*	BCL-6 co-repressor-like 1	NM_005094.3
2.329	*SSTR5*	Somatostatin receptor 5	NM_001053.1
2.173	*HOXD10*	Homebox D10	NM_014212.4
2.083	*HSPA1B*	Heat shock 70 kDa protein 1B	NM_002148.3
2.003	*DEK*	DEK oncogene	NM_003472.2
0.490	*AKT2*	V-akt murine thymoma viral oncogene homologue 2	NM_001626.2
0.442	*SDC2*	Syndecan-2	NM_024424.5
0.393	*UCP3*	Uncoupling protein 3	NM_012068.5
0.450	*NPTXR*	Neuronal pentraxin receptor	NM_014293
0.382	*BCAN*	Brevican	NM_021948.3
0.176	*SOX4*	SRY (sex determining region Y)-box4	NM_006632.3

Fold change compared to HT168-M1 cells using the TaqMan assay.

**Table 5 ijms-23-02704-t005:** Expression of IFN resistance DEGs in metastatic melanoma tissues from the TCGA database treated with IFN.

Gene	Fold Change *	*p* Value	q Value
*WFDC1*	0.083	0.017	0.017
*BCAN*	13.458	0.041	0.043
*SOX4*	3.174	0.005	0.005
*RPE65*	25.802	0.021	0.021
*HOXC11*	15.213	0.077	0.055

* Ratio of the geometric mean of gene expression intensities of IFN (*n* = 27) versus any other (*n* = 6) treated groups. *p* value: two sample t-test, q value: FDR-adjusted *p* value.

**Table 6 ijms-23-02704-t006:** Testing the predictive role of IFN-resistance DEGs on interferon therapy of melanoma patients.

Gene	Fold Change *	*p* Value	q Value
*HSPB7*	0.178	0.024	0.057
*MT2A*	0.133	0.044	0.143
*HSF1*	0.452	0.047	0.029
*WFDC1*	0.051	0.047	0.057
*TPD52L1*	0.178	0.051	0.143
*DOCK11*	2.087	0.064	0.086
*DEK*	2.231	0.067	0.143
*SOX4*	2.522	0.072	0.114

* Ratio of the geometric mean of intensities of samples with PD/SD (*n* = 4) versus CR/PR (*n* = 4) for IFN therapy. *p* value: two sample t-test, q value: FDR-adjusted *p* value.

**Table 7 ijms-23-02704-t007:** Analysis of the predictive power of the components of the four most upregulated and four most downregulated DEGs in immune checkpoint inhibitor treated melanoma patients in vitro.

Gene	Non-Responders	Responders	*p* Value
*WFDC1*	437.8 (*n* = 180)	235.7 (*n* = 113)	0.0033
*GAGE2C*	1.38 (*n* = 13)	1.33 (*n* = 15)	0.78
*EFNA3*	116.1 (*n* = 180)	74.2 (*n* = 113)	2.606 × 10^−6^
*DDX10*	694.3 (*n* = 180)	882.2 (*n* = 113)	0.000253
*CAMK1*	228.5 (*n* = 180)	257.1 (*n* = 113)	0.0769
*PTBP1*	4446.9 (*n* = 180)	3688.1 (*n* = 113)	0.00004038
*TNFSF10*	1624.4 (*n* = 180)	1330.3 (*n* = 113)	0.29
*SLC17A3*	12.3 (*n* = 180)	11.8 (*n* = 113)	0.17

**Table 8 ijms-23-02704-t008:** Analysis of the predictive power of the components of the in vitro selected DEGs (Table 4) in immune checkpoint inhibitor treated melanoma patients.

Gene	Non-Responders	Responders	*p* Value
*AKT2*	2706 (*n* = 180)	3421 (*n* = 113)	7.98 × 10^−6^
*BCAN*	4363 (*n* = 180)	1260 (*n* = 113)	0.0282
*NPTXR*	272 (*n* = 180)	131 (*n* = 113)	0.00114
*SDC2*	2923 (*n* = 180)	1495 (*n* = 113)	1.72 × 10^−7^
*SOX4*	2404 (*n* = 180)	1075 (*n* = 113)	1.57 × 10^−11^
*UCP3*	44.5 (*n* = 180)	42.3 (*n* = 113)	0.842
*BCORL1*	486 (*n* = 180)	655 (*n* = 113)	0.000275
*DEK*	5941 (*n* = 180)	4201 (*n* = 113)	1.66 × 10^−6^
*EFHD1*	1245 (*n* = 180)	175 (*n* = 113)	2.21 × 10^−9^
*HOXD10*	55 (*n* = 180)	54.7 (*n* = 113)	0.31
*PDE6A*	28.2 (*n* = 180)	24.2 (*n* = 113)	0.0255
*HSPA1B*	7423 (*n* = 180)	13037 (*n* = 113)	0.000296
*SSTR5*	2.7 (*n* = 180)	2.2 (*n* = 113)	0.0345

*p* value tested by Mann−Whitney U test. Data are expressed as normalized gene expressions.

**Table 9 ijms-23-02704-t009:** Analysis of the predictive power of the components of the in vivo selected DEGs in immune checkpoint inhibitor treated melanoma patients.

Gene	Non-Responders	Responders	*p* Value
AQP1	3969.9 (*n* = 180)	1920.5 (*n* = 113)	2.13 × 10^−6^
CDKL3	73.4 (*n* = 180)	72.1 (*n* = 113)	0.11
IFI27	4437.5 (*n* = 180)	3544.7 (*n* = 113)	0.42
CDCA4	763.4 (*n* = 180)	372.3 (*n* = 113)	5.9 × 10^−9^

*p* value was tested by Mann-Whitney U test. Data are expressed as normalized gene expression.

## Data Availability

Original microarray data are submitted as Appendix A.

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
