# Peer review of "Identification of a Tumor Cell Associated Type I IFN Resistance Gene Expression Signature of Human Melanoma, the Components of Which Have a Predictive Potential for Immunotherapy"

_ijms, 2022, doi:10.3390/ijms23052704_

Round 1

Reviewer 1 Report

Ladanyi et al. report of a comparative transcriptome analysis of HT186 versus IFN2a resistant HT168-M1Res cell line in vitro and in vivo.  The study is well laid out and the transcriptome analysis is appropriate and the statistical analysis is fitting for the aims of the study.

The utility of the findings, in their present form is very limited and additional experiments would strengthen the impact of the study.

  1. The 13-gene in vitro stable core and the 4 gene in vivo stable core should be examined in at least two other IFN2a resistant human melanoma cell line among in vitro conditions to validate and generalize the findings – ideally using PCR analysis.
  2. If possible, using patient materials, the in vivo stable core should be evaluated either experimentally or on other databases that have not yet been searched.

Author Response

Responses, to Reviewer 1

We thank the Reviewer for analyzing our data and the manuscript and for constructive suggestions for improvement. We corrected the manuscript as much as we could and included some novel data based on the suggestions.  Please find here also our responses for suggestions.

  1. Please repeat the GES analysis on at least 2 other IFN-resistant cell lines.

Unfortunately, we are unable to perform such an analysis within the provided time frame. The development of new cell lines in vitro takes several months, their testing in vivo as xenografts for resistance  also takes 4-6 weeks, accordingly to establish novel models can be only a project of another half a year especially with PCR testing. However, we feel that the clinical validation can be more important than another experimental series.

  1. Using patient materials the observed GES or their components can be tested experimentally or in other databases.

Yes, we fully agree with the Reviewer that that must be the next step. To improve this manuscript, we have looked for in silico databases and (since the ICI-treated patients database included in our manuscript covered all available datasets), found only useful a small cohort of IFN-treated melanomas in TCGA: a primary tumor-based cohort of 33 patients and a metastatic melanoma cohort of 75 pts. We have run the test of our IFN-resistance GES on these cohorts, and the data are now included into the Results (see new Table S5, and Table 5-6), as well as Materials and Methods... ) Our data indicates a differential expression of WFDC1, BCAN and SOX4, RPE65 and MAPT in IFN treated metastatic melanomas. Although the clinical outcome data are available only for a limited set of IFN-treated metastatic melanoma patients, in this small cohort we found as predictive marker of clinical response, HSBP7, MT2A, HSF1 and WFDC1 our IFN-res GES. These data support our notion that the new GES contains IFN-resistance genes of general importance. This notion is also supported by the data on the ICI-treated melanoma cohort, since several members of the 79-IFN-res GES (17 actually) have predictive power for clinical response. Interestingly, the IFN treatment associated genes and the ICI-therapy predictive genes overlap by three genes: WFDC1, BCAN and MT2A, further supporting our view of a connection between the two biological processes.

Unfortunately the size of the TCGA IFN-treated cohort is quite limited, which does not allow a much detailed analysis of these issues. Accordingly, we have collected an IFN treated metastatic human melanoma biobank with complete clinical data where some of the questions raised here can be analyzed by a comprehensive genomic study, which is ongoing and would be the basis of another independent manuscript. 

Reviewer 2 Report

Dear Editor, thank you so much for inviting me to revise this manuscript.

This study addresses a current topic.

The manuscript is quite well written and organized. English could be improved.

Figures and tables are comprehensive and clear.

The introduction explains in a clear and coherent manner the background of this study.

We suggest the following modifications:

  • Introduction section: although the authors correctly included important papers in this setting, we believe some studies regarding biomarkers of response to immunotherapy should be cited within the introduction (PMID: 33535621 ; PMID: 33549983  ), only for a matter of consistency. We think it might be useful to introduce the topic of this interesting study.
  • Methods and Statistical Analysis: nothing to add.
  • Discussion section: Very interesting and timely discussion. Of note, the authors should expand the Discussion section, including a more personal perspective to reflect on. For example, they could answer the following questions – in order to facilitate the understanding of this complex topic to readers: what potential does this study hold? What are the knowledge gaps and how do researchers tackle them? How do you see this area unfolding in the next 5 years? We think it would be extremely interesting for the readers.

However, we think the authors should be acknowledged for their work. In fact, they correctly addressed an important topic, the methods sound good and their discussion is well balanced.

One additional little flaw: the authors could better explain the limitations of their work, in the last part of the Discussion.

We believe this article is suitable for publication in the journal although some revisions are needed. The main strengths of this paper are that it addresses an interesting and very timely question and provides a clear answer, with some limitations.

We suggest a linguistic revision and the addition of some references for a matter of consistency. Moreover, the authors should better clarify some points.

Author Response

Responses to Reviewer 2

We appreciate the constructive and positive opinion of the reviewer about our work. We corrected the manuscript according to the suggestions of the reviewer as follows.

  1. Biomarkers of ICI therapy are missing or incomplete in the Introduction.

We have expanded the Introduction (last paragraph) with biomarkers of ICI therapy of melanoma commenting on PD-L1 expression (controversial, new reference), TMB high status (as a positive predictive factor, new reference) and MSI status (a rare genetic condition in melanoma, due to which there are not much clinical data available concerning ICI efficacy; new reference). We have also completed the INTRO with novel genetic markers of ICI therapy like IFN and defensing homozygous loss. (new ref)

  1. Discussion issues
  2. What are the knowledge gaps?

There are limited data in the literature on the role of type I IFN resistance of melanoma, most of the studies dealt with IFN-g/type II issues. Although type I IFN was used and still can be used in melanoma therapy, there are limited data how this may affect genetic progression or the ICI therapy sensitivity. Equally important is the fact that type I IFN can also be produced by cells of the TME and less is known how this may affect melanoma progression.

  1. What potential this study holds?

We have shown here that type I IFN resistance involve disregulation of several IRG genes but interestingly the majority of the resistance signature belong to non-IRGs indicating that type I IFN resistance does not simply depend on IFN signaling but on other molecular pathways of melanoma. Secondly, and perhaps more importantly, certain elements of the type I IFN resistance signature are suggested to serve as predictive markers for ICI therapy. Since the primary aim of this study was not to discover novel predictive markers of ICI therapy of melanoma, these data may trigger more focused analysis of this question. The overlap between the IFN and ICI therapy resistance signatures validated on clinical samples (BCAN, SOX4 and WFDC1) are clearly support our notion and is highlighted in the Discussion.

  1. Future perspectives?

The biomarkers of ICI therapy of melanoma are limited today for TMB basically, since the role of PD-L1 is controversial, MSI status is a rare event (1.5%) so novel predictive markers are desperately needed. Even the characterization of the composition of the TME as predictive marker is still in infancy in melanoma unlike in other solid tumors. Accordingly, we may help to turn the attention to type I IFN signaling as a potential novel predictive marker of ICI therapy in melanoma.

  1. Limitations

The limitations of this study are that it is based on an experimental human melanoma model with all the characteristics of cell line dependency. However, our observation on the potential role of the elements of this experimental signature as biomarker in IFN-treated melanoma of the TCGA, (novel T5-6) or in the ICI-treated in silico cohort of melanomas  (T7-8) all suggest that type I IFN resistance GES may not be so restricted to this model. Unfortunately the size of the TCGA IFN-treated cohort is quite limited, which does not allow a much detailed analysis of these issues. Accordingly, we have collected an IFN treated metastatic human melanoma biobank with complete clinical data where some of the questions raised here can be analyzed by a comprehensive genomic study.

Round 2

Reviewer 2 Report

Acceptance.